# Sample Constrained Treatment Effect Estimation

**Raghavendra Addanki**
Adobe Research
raddanki@adobe.com

**David Arbour**
Adobe Research
darbour26@gmail.com

**Tung Mai**
Adobe Research
tumai@adobe.com

**Cameron Musco**
University of Massachusetts Amherst
cmusco@cs.umass.edu

**Anup Rao**
Adobe Research
anuprao@adobe.com

## Abstract

Treatment effect estimation is a fundamental problem in causal inference. We focus on designing efficient randomized controlled trials, to accurately estimate the effect of some treatment on a population of $n$ individuals. In particular, we study *sample-constrained treatment effect estimation*, where we must select a subset of $s \ll n$ individuals from the population to experiment on. This subset must be further partitioned into treatment and control groups. Algorithms for partitioning the entire population into treatment and control groups, or for choosing a single representative subset, have been well-studied. The key challenge in our setting is jointly choosing a representative subset and a partition for that set.

We focus on both individual and average treatment effect estimation, under a linear effects model. We give provably efficient experimental designs and corresponding estimators, by identifying connections to discrepancy minimization and leverage-score-based sampling used in randomized numerical linear algebra. Our theoretical results obtain a smooth transition to known guarantees when $s$ equals the population size. We also empirically demonstrate the performance of our algorithms.

## 1 Introduction

Experimentation has long been held as a gold standard for inferring causal effects since one can explicitly enforce independence between treatment assignment and other variables which influence the outcome of interest. We consider the potential outcomes framework [37, 40], where each individual is associated with a control and treatment value (also called the *potential outcomes*) and based on the treatment assignment, we can observe only one of these values. Efficient designs of experimentation for estimating individual treatment effects which measure the difference between treatment and control values for each individual, and the average treatment effect which measures the average individual treatment effect has been well-studied [35]. In the absence of assumptions on the functional form of the potential outcomes, the minimax optimal approach for conducting an experiment is to assign individuals to treatment or control completely at random, without consideration of baseline covariates (features) [25]. However, by considering covariates for each individual, and using additional assumptions of smoothness, substantial gains can be made in terms of the variance of the treatment effect estimate via alternative assignment procedures. The most common approach attempts to minimize imbalance, i.e., the difference between the baseline covariates in the treatment and control groups [6, 25, 35].

While experimental designs that minimize imbalance increase the power of an experiment for a given pool of subjects, there are many practical applications where the experimenter wishes to minimize

---

Author ordering is alphabetical.

36th Conference on Neural Information Processing Systems (NeurIPS 2022).

the total number of subjects who are placed into the experiment. For example, in medicine, clinical trials may carry nontrivial risk to patients. Within industrial applications, experiments may carry substantial costs in terms of testing changes, which decrease the quality of the user experience, or have direct monetary costs.

In this paper, we examine the problem of selecting a subset of $s$ individuals from a larger population and assigning treatments such that the estimated treatment effect has a small error. We consider two different estimands: individual treatment effect (ITE) and average treatment effect (ATE).

A bit more formally, we represent the $d$-covariates of a population of $n$ individuals using $\mathbf{X} \in \mathbb{R}^{n \times d}$. We assume that the treatment and control values, denoted by $\mathbf{y}^1, \mathbf{y}^0 \in \mathbb{R}^n$, are functions of the covariates, i.e., $\mathbf{y}^1 = f(\mathbf{X}, \boldsymbol{\zeta}^1)$ and $\mathbf{y}^0 = g(\mathbf{X}, \boldsymbol{\zeta}^0)$ where $\boldsymbol{\zeta}^0, \boldsymbol{\zeta}^1 \in \mathbb{R}^n$ are noise vectors. The ITE for the $i^{th}$ individual is $\mathbf{y}_i^1 - \mathbf{y}_i^0$ and ATE is the average of all the ITE values. We further assume a linear model, i.e., the functions $f, g$ are linear in $\mathbf{X}$ and $\boldsymbol{\zeta}^1, \boldsymbol{\zeta}^0$. The goal is to pick a subset of $s$ individuals and partition this subset into control and treatment groups. For an individual $i$ in the treatment group, we measure $\mathbf{y}_i^1$, and for an individual $j$ in the control, we measure $\mathbf{y}_j^0$. From this small set of measurements, we seek to estimate the ITE or ATE over the full population.

Without parametric assumptions, ITE estimation is not feasible [43]. We focus on linear models in particular, since they are important in developing theory. E.g., in the literature on optimal designs in active learning, much of the foundational theory is built around linear models. Identifying estimators based on linearity assumptions is an active area of study in the causal inference literature [20, 50].

Our setup is similar to active learning [42], where the goal is to minimize the number of individual labels that we access for solving linear regression or other downstream tasks. The key difference is that we must select both a subset of individuals, and for each $i$, can measure only one of two labels: $\mathbf{y}_i^1$ or $\mathbf{y}_i^0$. In particular, ITE estimation can be thought of as solving two *simultaneous* active linear regression problems – one for the treatment outcomes and one for the control outcomes. Thus, standard active learning-based approaches, such as [11, 12, 34], fall short. Even when $s$ equals the population size $n$, i.e., when active learning becomes trivial, our problem does not. We must still pick a partition of the full population into treatment and control groups. Overall, sample constrained treatment effect estimation by designing efficient randomized controlled trials has received little attention, compared to various approaches that use observational data, such as [24, 39, 46].

**Our Contributions.** For ITE estimation, we propose an algorithm using *leverage score sampling* [51], which is a popular approach to subset selection for fast linear algebraic computation. For ATE estimation, we employ a recursive application of a covariate balancing design [20]. We provide a theoretical analysis in terms of root mean squared error (ITE) and deviation error (ATE).

Recall that we assume the treatment and control values are linear functions of the covariates plus Gaussian noise, i.e., $\mathbf{y}^1 = \mathbf{X}\boldsymbol{\beta}^1 + \boldsymbol{\zeta}^1$ and $\mathbf{y}^0 = \mathbf{X}\boldsymbol{\beta}^0 + \boldsymbol{\zeta}^0$ where $\boldsymbol{\zeta}^1, \boldsymbol{\zeta}^0 \in \mathbb{R}^n$ have i.i.d. mean zero, variance $\sigma^2$ Gaussian entries, and $\boldsymbol{\beta}^1, \boldsymbol{\beta}^0 \in \mathbb{R}^d$ are coefficient vectors.

For ITE estimation, we give a randomized algorithm that selects $\Theta(d \log d)$ individuals in expectation, using leverage scores, which measure the importance of an individual based on their covariates. Our algorithm obtains, with high probability, root mean squared error $O\left(\sqrt{\log d / n} \cdot (\|\boldsymbol{\beta}^1\| + \|\boldsymbol{\beta}^0\|) + \sigma\right)$ (see Corollary 3.7). We argue that this is optimal up to constants and a $\sqrt{\log d}$ factor, *even for approaches that experiment on the full population.*

The key challenge in achieving this bound is to extend leverage scores to our simultaneous linear regression setting, ensuring that we do not share samples across the treatment and control effect estimation problems. To do this, we introduce a *smoothed* covariate matrix, whose leverage scores are bounded. This ensures that, when applying independent leverage score sampling, with high probability few individuals are randomly assigned to both control and treatment, and thus removing such individuals from one of the groups does not introduce too much error.

For ATE estimation we give a randomized algorithm that selects at most $s$ individuals for treatment/control assignment and obtains an error of $\widetilde{O}\left(\sigma/\sqrt{s} + (\|\boldsymbol{\beta}^1\| + \|\boldsymbol{\beta}^0\|)/s\right)$, where $\widetilde{O}(\cdot)$ hides logarithmic factors (see Theorem 4.3). The error decreases with increasing values of $s$ and when $s = n$, it matches state-of-the-art guarantees due to Harshaw et al. [20].

Our algorithm for ATE estimation is based on *covariate balancing*. This is a popular approach where one attempts to assign similar individuals to the treatment and control groups, to ensure that the

observed effect is attributed to the administered treatment alone. Harshaw et al. [20] designed an algorithm by minimizing the discrepancy of an augmented covariate matrix, which achieves low ATE estimation error. To extend their approach to our setting, first, we need to select a subset of $s$ individuals that are representative of the entire population, and then balance the covariates. Uniform sampling or importance sampling techniques give high error here. Instead, we employ a recursive strategy, which repeatedly partitions the individuals into two subsets by balancing covariates, and selects the smaller subset to recurse on, until we have selected at most $s$ individuals.

We observe that our techniques for ITE and ATE estimation should extend to the setting when the outcomes are non-linear functions of the covariates, which are linear in some higher-dimensional kernel space. This is immediate for our discrepancy minimization design for ATE, which only requires knowing the pairwise inner products of the covariate vectors. For ITE estimation, leverage score sampling for kernel ridge regression [3] is most likely applicable. Extensions to broader classes of non-linear models are beyond the scope of this work, but they are an interesting future direction.

Finally, in Section 5, we provide an empirical evaluation of the performance of our ITE and ATE estimation methods, comparing against uniform sampling and other baselines on several datasets. Our results suggest that our techniques can help reduce the costs associated with running a randomized controlled trials substantially using only a small fraction of the population.

**Other Related Work.** For ATE estimation, the most well-studied approaches to experiment design are covariate balancing and randomization. A variety of design techniques have been studied based on these approaches, such as blocking [18], matching [23, 45], rerandomization [30, 35], and optimization [25]. Using observational data, treatment effect estimation using covariate regression adjustment [31] and various active learning-based sampling techniques have gained recent attention [24, 38, 46]. Compared to ATE, estimating ITE is significantly harder and has received attention only recently using machine learning methods [7, 43, 49]. There has been a lot of recent work on efficient experimental designs to minimize experimental costs, in various domains, such as causal discovery [1, 2, 16, 17, 28, 44], multi-arm bandits [4, 27, 36], and group testing [9, 10, 15].

## 2 Preliminaries

**Notation.** We use bold capital letters, e.g., $\mathbf{X}$ to denote matrices and bold lowercase letters, e.g., $\mathbf{y}$ to denote vectors. We use $\mathbf{X}[i, :]$ and $\mathbf{X}[:, j]$ to denote the $i^{th}$ row and $j^{th}$ column of $\mathbf{X}$ respectively, which we always view as column vectors. The $i^{th}$ largest singular value of $\mathbf{X}$ is denoted by $\sigma_i(\mathbf{X})$. For any vector $\mathbf{x}$, the Euclidean norm or the $\ell_2$-norm is denoted by $\|\mathbf{x}\|$.

For a population of $n$ individuals, we represent each with an integer in $[n]$ where we denote $[n] \overset{\text{def}}{=} \{1, 2, \cdots, n\}$. Each individual $j \in [n]$ is associated with a treatment and a control value, denoted $\mathbf{y}_j^1, \mathbf{y}_j^0 \in \mathbb{R}^+$, respectively. The vectors associated with all $n$ treatment and control values are denoted $\mathbf{y}^1$ and $\mathbf{y}^0$. Additionally, each individual is associated with a $d$-dimensional covariate vector. Combined, they comprise the rows of the covariate matrix $\mathbf{X} \in \mathbb{R}^{n \times d}$.

In this paper, we consider the finite population framework, where the potential outcomes of individuals are fixed and the randomness is only due to treatment assignment [13]. We make the SUTVA assumption, i.e., the treatment outcome value of any individual is independent of treatment assignments of others in the population [48].

**Assumption 2.1** (Linearity Assumption). *Under the linearity assumption, the treatment and control values are a linear function of the covariates. Formally, for some $\boldsymbol{\beta}^0, \boldsymbol{\beta}^1 \in \mathbb{R}^d$,*

$$\mathbf{y}^1 = \mathbf{X}\boldsymbol{\beta}^1 + \boldsymbol{\zeta}^1 \text{ and } \mathbf{y}^0 = \mathbf{X}\boldsymbol{\beta}^0 + \boldsymbol{\zeta}^0,$$

*where $\boldsymbol{\zeta}^1, \boldsymbol{\zeta}^0 \in \mathbb{R}^n$ are noise vectors, with each coordinate drawn independently from the Gaussian distribution with zero mean and variance $\sigma^2$, i.e., $N(0, \sigma^2)$. We further assume that $\mathbf{X}$ is row-normalized, i.e., $\|\mathbf{X}[i, :]\| \leq 1 \, \forall i \in [n]$.*

**Definition 2.2** (Individual Treatment Effect). *Given a population of $n$ individuals, the individual treatment effect (ITE) of $j \in [n]$ is the difference between the treatment and control values:*

$$\text{ITE}(j) \overset{\text{def}}{=} \mathbf{y}_j^1 - \mathbf{y}_j^0.$$

**Definition 2.3** (Average Treatment Effect). *Given a population of $n$ individuals, the average treatment effect (ATE), denoted by $\tau$, is the average individual treatment effect:*

$$\tau \overset{\text{def}}{=} \frac{1}{n} \sum_{j \in [n]} \text{ITE}(j) = \frac{1}{n} \sum_{j \in [n]} \mathbf{y}_j^1 - \mathbf{y}_j^0.$$

**Definition 2.4** (Root Mean Squared Error). *For a set of estimated individual treatment effects, $\widehat{\text{ITE}}(j)$ for $j \in [n]$, the root mean squared error (RMSE) is defined as:*

$$\text{RMSE} \overset{\text{def}}{=} \frac{1}{\sqrt{n}} \cdot \left\| \widehat{\text{ITE}}(j) - \text{ITE}(j) \right\|.$$

**Definition 2.5** (Leverage Score). *Given a matrix $\mathbf{X} \in \mathbb{R}^{n \times d}$, the leverage score of $j^{th}$ row $\mathbf{X}[j, :]$, denoted by $\ell_j(\mathbf{X})$, is defined as:*

$$\ell_j(\mathbf{X}) \overset{\text{def}}{=} \mathbf{X}[j, :]^T (\mathbf{X}^T \mathbf{X})^+ \mathbf{X}[j, :],$$

*where $^+$ denotes the Moore–Penrose pseudo-inverse.*

## 3 Individual Treatment Effect Estimation

We now describe our algorithm for ITE estimation. The algorithm identifies a subset of the population to experiment on, using *importance based sampling* techniques, that are well-studied in randomized numerical linear algebra [51]. Missing proof details in this section are presented in Appendix A.1.

**Overview of our approach.** Under the linearity assumption (Assumption 2.1), we can reformulate the problem of estimating the ITE for every individual as simultaneously solving two linear regression instances: one for control and one for treatment, i.e., we regress $\mathbf{y}^0, \mathbf{y}^1$ on $\mathbf{X}$. However, there are two challenges: 1) we would like to solve these regression problems using measurements from just a small subset of $s$ individuals and 2) we only have access to either the control or treatment measurement $\mathbf{y}_j^0$ or $\mathbf{y}_j^1$ for any individual in this set.

To tackle the first challenge, we use a sampling technique based on the importance of each row in $\mathbf{X}$, captured via its leverage score (Defn. 2.5). Intuitively, we want to select $s$ individuals (or equivalently rows) that capture the entire row space of $\mathbf{X}$ and use them to estimate the ITE of all other individuals. Leverage scores capture the importance of a row in making up the row space. E.g., if a row is orthogonal to all the other rows, it's leverage score will be the maximum value of $1$.

Unfortunately, if we apply leverage score sampling independently to the regression problems for $\mathbf{y}^0$ and $\mathbf{y}^1$, rows with high leverage leverage scores may be sampled for both instances. This presents a problem, since we can only read at most one of $\mathbf{y}_j^0$ or $\mathbf{y}_j^1$. To mitigate this issue, we construct a *smoothed* matrix $\mathbf{X}^*$, which consists of $\mathbf{X}$ projected onto its singular vectors with high singular values. Intuitively, this dampens the effects of high leverage score 'outlier' rows that don't contribute significantly to the spectrum of $\mathbf{X}$. Formally, we prove that the maximum leverage score of $\mathbf{X}^*$ is bounded, which let's us solve our two regression problems via independent sampling. There will be few repeated samples across our subsets, which introduce minimal error.

### 3.1 Leverage Score Sampling

For some $\gamma \geq 0$, to be fixed later, we define a *smoothed* matrix for $\mathbf{X}$, the projection onto singular vectors with high singular values, as follows:

**Definition 3.1** (Smoothed matrix). *Given $\mathbf{X} \in \mathbb{R}^{n \times d}$ with singular value decomposition $\mathbf{X} = \mathbf{U}\mathbf{\Sigma}\mathbf{V}^T$, let $\Gamma^*$ be the set of indices corresponding to singular values greater than $\sqrt{\gamma}$, i.e., $\Gamma^* \overset{\text{def}}{=} \{i \mid \sigma_i(\mathbf{X}) \geq \sqrt{\gamma}\}$; we denote $d' \overset{\text{def}}{=} |\Gamma^*|$. Let $\mathbf{\Sigma}^* = \mathbf{\Sigma}(\Gamma^*, \Gamma^*)$ denote the principal sub-matrix of $\mathbf{\Sigma}$ associated with these large singular values. Similarly, let $\mathbf{U}^* \in \mathbb{R}^{n \times d'}, \mathbf{V}^* \in \mathbb{R}^{d \times d'}$ be the associated column sub-matrices of $\mathbf{U}$ and $\mathbf{V}$. Then, we define:*

$$\mathbf{X}^* \overset{\text{def}}{=} \mathbf{U}^* \mathbf{\Sigma}^* \mathbf{V}^{*T}.$$

**Sampling Matrix.** Our algorithm will sample individuals, corresponding to rows of the smoothed matrix of $\mathbf{X}$, i.e., $\mathbf{X}^*$, independently – the $i^{th}$ row is included in the sample with some probability $\boldsymbol{\pi}_i$. Let the set of rows sampled be denoted by $S$. We can associate a sampling matrix $\mathbf{W}$ with $S$. The $j^{th}$ row of $\mathbf{W}$ is associated with the $j^{th}$ element in the set $S$ (under some fixed order). If the $j^{th}$ element in $S$ is the row for individual $i$ for some $i \in [n]$, then, $\mathbf{W}[j,:]$ is equal to $\mathbf{e}_i/\sqrt{\boldsymbol{\pi}_i}$. Here, $\mathbf{e}_i \in \mathbb{R}^n$ denotes the $i^{th}$ standard basis vector. In this way, $\mathbf{W}\mathbf{X}^*$ consists of the subset of rows sampled in $S$, reweighted by the inverse squareroot of their sampling probabilities, which is necessary to keep expectations correct in solving the linear regression.

---

**Algorithm 1** SAMPLING-ITE

---

    **Input:** Smoothed covariates $\mathbf{X}^* \in \mathbb{R}^{n \times d}$, sampling probabilities $\boldsymbol{\pi} \in [0,1]^n$.
    **Output:** Estimates for ITE$(j)$ for each individual $j \in [n]$.
1: Add each $j \in [n]$ to set $S^0$ independently, with prob. $\boldsymbol{\pi}_j$.
2: Add each $j \in [n]$ to set $S^1$ independently, with prob. $\boldsymbol{\pi}_j$.
3: Construct sampling matrix $\mathbf{W}^0$ from $S^0$ using probabilities $\boldsymbol{\pi}$.
4: Construct sampling matrix $\mathbf{W}^1$ from $S^1 \setminus S^0$ using probabilities $\boldsymbol{\pi}(1-\boldsymbol{\pi})$.
5: Let $\widetilde{\boldsymbol{\beta}}^i = \arg\min_{\boldsymbol{\beta} \in \mathbb{R}^d} \left\| \mathbf{W}^i \mathbf{X}^* \boldsymbol{\beta} - \mathbf{W}^i \mathbf{y}^i \right\|^2$ for $i = 0, 1$.
6: For each $j \in [n]$, let $\widehat{\text{ITE}}(j)$ be the $j^{th}$ entry of the vector $\mathbf{X}^* \widetilde{\boldsymbol{\beta}}^1 - \mathbf{X}^* \widetilde{\boldsymbol{\beta}}^0$ .
7: **return** $\widehat{\text{ITE}}(j)$ $\forall j \in [n]$.

---

**Algorithm SAMPLING-ITE.** We perform row sampling twice, with probabilities proportional to the leverage scores of $\mathbf{X}^*$, to construct two sets $S^0, S^1$. See the discussion below for the exact definition of the sampling probabilities $\boldsymbol{\pi}_i$, which are proportional to the leverage scores of $\mathbf{X}^*$. These two sets are used to estimate the vectors $\mathbf{y}^0$ and $\mathbf{y}^1$, respectively. It is possible that a row gets included in both $S^0$ and $S^1$. In that case, we simply remove the row from $S^1$. As a result, $j^{th}$ row is included in $S^1$ with probability $\boldsymbol{\pi}_j \cdot (1 - \boldsymbol{\pi}_j)$ for every $j \in [n]$. We construct sampling matrices $\mathbf{W}^0$ and $\mathbf{W}^1$ using probabilities $\boldsymbol{\pi}$ and $\boldsymbol{\pi}(1 - \boldsymbol{\pi})$ respectively. Finally, in Algorithm 1, we solve the following linear regressions, for $i = 0, 1$ separately:

$$\widetilde{\boldsymbol{\beta}}^i = \arg\min_{\boldsymbol{\beta} \in \mathbb{R}^d} \left\| \mathbf{W}^i \mathbf{X}^* \boldsymbol{\beta} - \mathbf{W}^i \mathbf{y}^i \right\|^2$$

Our estimate for each ITE$(j)$, denoted by $\widehat{\text{ITE}}(j)$ is set to $j^{th}$ entry of the vector $\mathbf{X}^* \widetilde{\boldsymbol{\beta}}^1 - \mathbf{X}^* \widetilde{\boldsymbol{\beta}}^0$. Observe that by construction, $S^0 \cap S^1$ is empty. This ensures that we have access to only one of $\mathbf{y}_j^0$ or $\mathbf{y}_j^1$ for any individual $j$ in solving the above two subsampled regression problems.

We note that in Algorithm 1, we could remove $j$ from one of $S^0, S^1$, or with equal probability from either of the two sets, and obtain the exact same guarantees.

### 3.2 Theoretical Guarantees

First, we bound the error due to sampling. Critically, we show that the leverage scores of $\mathbf{X}^*$, and in turn the probabilities $\boldsymbol{\pi}$, are bounded by $1/\gamma$. Thus, the sampling probabilities for $S^1$, $\boldsymbol{\pi}(1 - \boldsymbol{\pi})$ are not too far from $\boldsymbol{\pi}$ itself.

As we assume the row norms of $\mathbf{X}$ are bounded by 1, the row norms of $\mathbf{X}^*$ are also bounded. Thus, there can be no rows in $\mathbf{X}^*$ that are nearly orthogonal to all other rows – i.e., there can be no rows with very high leverage scores. Such rows would lead to small singular values. However, we know that the smallest singular value of $\mathbf{X}^*$ is at least $\sqrt{\gamma}$. In particular, we prove:

**Claim 3.2.** $\ell_j(\mathbf{X}^*) \leq 1/\gamma$, for all $j \in [n]$.

**Setting $\boldsymbol{\pi}$.** It is well known that if we sample rows of $\mathbf{X}^*$ with probabilities $\boldsymbol{\pi}$ proportional to the leverage scores, we will obtain a $(1 \pm \epsilon)$ relative error approximation for linear regression [41]. The result of Sarlos [41] applies to sampling $s$ rows *with replacement*, each equal to $j$ with probability $\boldsymbol{\pi}_j / \|\boldsymbol{\pi}\|$. It is not hard to observe that it extends to the variant where each row is included in the sample independently with similar probability. Therefore, we have:

**Lemma 3.3** (Follows from [41])**.** *For $\mathbf{X} \in \mathbb{R}^{n \times d}$, $\mathbf{y} \in \mathbb{R}^n$, let $S \subseteq [n]$ include each $j \in [n]$ independently with probability $\boldsymbol{\pi}_j$ satisfying $\boldsymbol{\pi}_j \geq \min\left\{1, \ell_j(\mathbf{X}) \cdot c \cdot [\log(\text{rank}(\mathbf{X})) + \frac{1}{\delta\epsilon}]\right\}$ for*

*some large enough constant c. Let $\mathbf{W} \in \mathbb{R}^{|S| \times n}$ be a sampling matrix that includes row $\mathbf{e}_j/\sqrt{\boldsymbol{\pi}_j}$ if $j \in S$, where $\mathbf{e}_j \in \mathbb{R}^n$ is the $j^{th}$ standard basis vector. Let $\widetilde{\boldsymbol{\beta}} = \arg\min_{\boldsymbol{\beta} \in \mathbb{R}^d} \|\mathbf{W}\mathbf{X}\boldsymbol{\beta} - \mathbf{W}\mathbf{y}\|^2$. Then, $\mathbb{E}[|S|] = \sum_{j=1}^n \boldsymbol{\pi}_j$ and with probability $\geq 1 - \delta$:*

$$\left\|\mathbf{X}\widetilde{\boldsymbol{\beta}} - \mathbf{y}\right\| \leq (1 + \epsilon) \cdot \min_{\boldsymbol{\beta}} \|\mathbf{X}\boldsymbol{\beta} - \mathbf{y}\|.$$

*If the $\boldsymbol{\pi}_j$'s are within constants of the required bound, $\mathbb{E}[|S|] = O\left(d \log d + \frac{d}{\epsilon \delta}\right)$.*

Note that the bound on $\mathbb{E}[|S|]$ follows from the well known fact that the sum of leverage scores, is equal to the rank, i.e., $\sum_{j=1}^n \ell_j(\mathbf{X}) = \text{rank}(\mathbf{X}) \leq d$ [51].

The sampling probabilities are set to $\boldsymbol{\pi}_j = \min\{1, \ell_j(\mathbf{X}^*) \cdot c_0 \cdot [\log(\text{rank}(\mathbf{X}^*)) + 30/\epsilon]\}$ for some constant $c_0 \geq 2c$, where $c$ is the constant in Lemma 3.3. Thus, by the lemma, we will have, with probability $\geq 29/30$, $\left\|\mathbf{X}^*\widetilde{\boldsymbol{\beta}}^0 - \mathbf{y}^0\right\| \leq (1 + \epsilon)\left\|\mathbf{X}^*\boldsymbol{\beta}^0 - \mathbf{y}^0\right\|$.

It remains to show that we will have a similar guarantee for the control group. The rows in $S^1$ are included independently with probability $\boldsymbol{\pi}_j \cdot (1 - \boldsymbol{\pi}_j)$. If we can prove that $\boldsymbol{\pi}_j \cdot (1 - \boldsymbol{\pi}_j) \geq \frac{\boldsymbol{\pi}_j}{2}$, then Lemma 3.3 will still apply, since we have set $c_0 = 2c$. To do so, it suffices to argue that $\boldsymbol{\pi}_j \leq 1/2$ by setting the parameters appropriately.

**Claim 3.4.** *If $\gamma = 4c_0 \max\{\log(\text{rank}(\mathbf{X}^*)), 30/\epsilon\}$ and $\boldsymbol{\pi}_j = \min\{1, \ell_j(\mathbf{X}^*) \cdot c_0 \cdot [\log(\text{rank}(\mathbf{X}^*)) + 30/\epsilon]\}$, we have $\boldsymbol{\pi}_j \leq 1/2$ for every $j \in [n]$.*

*Proof.*

$$\boldsymbol{\pi}_j \leq \ell_j(\mathbf{X}^*) \cdot c_0 \cdot [\log(\text{rank}(\mathbf{X}^*)) + 30/\epsilon] \leq 1/\gamma \cdot c_0 \cdot [\log(\text{rank}(\mathbf{X}^*)) + 30/\epsilon] \quad \text{(Claim 3.2)}$$

$$\leq \frac{c_0[\log(\text{rank}(\mathbf{X}^*)) + 30/\epsilon]}{4c_0 \max\{\log(\text{rank}(\mathbf{X}^*)), 30/\epsilon\}} \leq \frac{1}{2}.$$

$\square$

In Appendix A.1, we argue that using the smoothed matrix $\mathbf{X}^*$ introduces an error of $\sqrt{\gamma}$. Combining all of them, we have the following corollary:

**Corollary 3.5.** *Suppose $\gamma$ and $\boldsymbol{\pi}_j$ are set as in Claim 3.4, for some sufficiently large constant $c_0$. Then, Algorithm SAMPLING-ITE satisfies, for $i = 0, 1$, with probability at least $14/15$:*

$$\left\|\mathbf{X}^*\widetilde{\boldsymbol{\beta}}^i - \mathbf{y}^i\right\| \leq (1 + \epsilon) \cdot \left(\sqrt{\gamma}\left\|\boldsymbol{\beta}^i\right\| + \left\|\boldsymbol{\zeta}^i\right\|\right) \text{ for } i = 0, 1.$$

**RMSE Guarantees.** The root mean squared error (Defn. 2.4) for the ITE estimates is given by:

$$\text{RMSE} = \frac{1}{\sqrt{n}}\left\|(\mathbf{X}^*\widetilde{\boldsymbol{\beta}}^1 - \mathbf{X}^*\widetilde{\boldsymbol{\beta}}^0) - (\mathbf{y}^1 - \mathbf{y}^0)\right\|.$$

By setting $\epsilon = 120c_0 d \log d/s$ in Corollary 3.5, we get the following theorem for our Algorithm 1:

**Theorem 3.6.** *Suppose $s \geq 120c_0 d \log d$. There is a randomized algorithm that selects a subset $S \subseteq [n]$ of the population with $\mathbf{E}[|S|] \leq s$, and, with probability at least $9/10$, returns ITE estimates $\widehat{\text{ITE}}(j)$ for all $j \in [n]$ with error:*

$$\text{RMSE} = O\left(\sqrt{\frac{1}{n}\max\left\{\frac{s}{d}, \log d\right\}} \cdot (\|\boldsymbol{\beta}^1\| + \|\boldsymbol{\beta}^0\|) + \sigma\right).$$

For the sake of simplicity of analysis, we used a constant success probability in Theorem 3.6. All our claims can easily be updated with a general failure probability of $\delta$, with a dependence of $\sqrt{1/\delta}$, using Lemma 3.3. The corollary below follows immediately from Theorem 3.6.

**Corollary 3.7** (Main ITE Error Bound). *The root mean squared error obtained by Algorithm 1 is minimized when $s = \Theta(d \log d)$ and is given by:*

$$\text{RMSE} = O\left(\sqrt{\frac{\log d}{n}} \cdot (\|\boldsymbol{\beta}^1\| + \|\boldsymbol{\beta}^0\|) + \sigma\right).$$

Our upper bound on RMSE for Algorithm 1 increases with $s$, if $s$ grows strictly faster than $d \log d$ asymptotically, i.e., $s = \omega(d \log d)$. Therefore, to obtain low error, we set $s = c \cdot d \log d$ for some constant $c$, even if the sample constraint allows for larger values. We believe this is an artifact of our analysis. In Section 5, we observe empirically that the error decreases with increasing $s$.

**Remark.** We observe that the RMSE bound in Corollary 3.7 is nearly optimal, even for algorithms that *experiment on the full population*. The $O(\sigma)$ term cannot be improved by more than constants, as a consequence of our noise model (see Assumption 2.1). Even if we knew the true $\boldsymbol{\beta}^1$ and $\boldsymbol{\beta}^0$, our RMSE would be $O(\sigma)$.

The term $(\|\boldsymbol{\beta}^0\| + \|\boldsymbol{\beta}^1\|)/\sqrt{n}$ is also necessary. Suppose the matrix $\mathbf{X}$ is such that all rows, except row $j$, are zero vectors. Row $j$ is a standard basis vector, i.e., its $i^{th}$ entry is 1 for some $i$. Suppose also that $\boldsymbol{\beta}^1$ and $\boldsymbol{\beta}^0$ are both independently set to the same standard basis vector with probability $1/2$, and set to zero otherwise. Then, with probability $1/2$, ITE$(j) = 0$ and with probability $1/2$, ITE$(j) = \pm 1$. No algorithm which observes just one of $\mathbf{y}_j^1$ or $\mathbf{y}_j^0$ can obtain expected error o(1) in estimating ITE$(j)$. That is, no algorithm can obtain RMSE $o(1/\sqrt{n}) = o\left((\|\boldsymbol{\beta}^0\| + \|\boldsymbol{\beta}^1\|)/\sqrt{n}\right)$.

# 4 Average Treatment Effect Estimation

In this section, we describe our approach for estimating the average treatment effect, under the sample constraint, by building upon a recent work on efficient experimental design by Harshaw et al. [20]. Missing details from this section are collected in Appendix A.2.

**Horvitz-Thompson Estimator.** Suppose $\mathbf{S}^+ \subseteq [n]$ is the population assigned to the treatment group and $\mathbf{S}^- = [n] \setminus \mathbf{S}^+$ is the remaining population, i.e., the control group. A well-studied estimator for estimating the average treatment effect is the Horvitz-Thompson estimator [22], denoted by $\widehat{\tau}$. If every individual is assigned to $\mathbf{S}^+$ (or $\mathbf{S}^-$) with probability 0.5, then, $\widehat{\tau}$ is defined as follows:

$$\widehat{\tau} = \frac{2}{n}\left(\sum_{i \in \mathbf{S}^+} \mathbf{y}_i^1 - \sum_{i \in \mathbf{S}^-} \mathbf{y}_i^0\right).$$

---

**Algorithm 2** RECURSIVE-COVARIATE-BALANCING

**Input:** Covariate matrix $\mathbf{X} \in \mathbb{R}^{n \times d}$, number of experiments to be run $s$.
**Output:** Estimate for ATE.
1: Set $t = 1$, $\mathbf{Z}_t := \mathbf{X}$, $n_t = n$.
2: **while True do**
3:     $\mathbf{Z}_t^+, \mathbf{Z}_t^- \leftarrow$ GRAM-SCHMIDT-WALK$(\mathbf{Z}_t, \delta')$ where $\delta' = \log(16 \log(n/s))$.
4:     **if** $n_t \leq s$ **then**
5:         **break**
6:     **else if** size$(\mathbf{Z}_t^+) \geq$ size$(\mathbf{Z}_t^-)$ **then**
7:         Set $\mathbf{Z}_{t+1} \leftarrow \mathbf{Z}_t^-$ and $n_{t+1} \leftarrow$ size$(\mathbf{Z}_t^-)$.
8:     **else**
9:         Set $\mathbf{Z}_{t+1} \leftarrow \mathbf{Z}_t^+$ and $n_{t+1} \leftarrow$ size$(\mathbf{Z}_t^+)$.
10:     **end if**
11:     $t \leftarrow t + 1$
12: **end while**
13: Use $\mathbf{Z}_t^+, \mathbf{Z}_t^-$ to construct the ATE estimator as: $\widehat{\tau}_s = 2^t/n \cdot \left(\sum_{j \in \mathbf{z}_t^+} \mathbf{y}_j^1 - \sum_{j \in \mathbf{z}_t^-} \mathbf{y}_j^0\right)$.
14: **return** $\widehat{\tau}_s$.

---

Harshaw et al. [20] present an experimental design based on the Gram-Schmidt-Walk algorithm for discrepancy minimization [8]. Their Gram-Schmidt-Walk design produces a random partition of the population with a good balance in every dimension, i.e., control and treatment groups have similar covariates. For the Horvitz-Thompson estimator, they give a tradeoff between covariate balancing and robustness (estimation error). Formally, they obtain:

**Lemma 4.1** (Proposition 3 in [20]). *For all* $\Delta > 0$, *with probability at least* $1 - 2\exp\left(-\frac{\Delta^2 n}{8L}\right)$, *the Gram-Schmidt-Walk design satisfies:* $|\widehat{\tau} - \tau| \leq \Delta$, *where* $L = \frac{2}{n}\min_{\boldsymbol{\beta} \in \mathbb{R}^d}\left(\left\|\frac{\mathbf{y}^1+\mathbf{y}^0}{2} - \mathbf{X}\boldsymbol{\beta}\right\|^2 + \|\boldsymbol{\beta}\|^2\right)$.

**Overview of RECURSIVE-COVARIATE-BALANCING.** Our main idea in Algorithm 2 is to partition the population using the Gram-Schmidt-Walk design (GSW) recursively until the total size of population that we can experiment on reduces to $s$. In each recursive call, we start by partitioning the available individuals $\mathbf{Z}_t$ into treatment and control groups, denoted by $\mathbf{Z}_t^+, \mathbf{Z}_t^-$ using GSW. Next, we identify the smaller of these two subsets, say $\mathbf{Z}_t^+$ and recurse on $\mathbf{Z}_t^+$. We stop after $k$ recursive calls when there are only $s$ individuals to experiment on, i.e., $|\mathbf{Z}_k^+ \cup \mathbf{Z}_k^-| \leq s$. Finally, we construct our estimator $\widehat{\tau}_s$, similar to the Horvitz-Thompson estimator, by scaling the treatment and control contributions due to $\mathbf{Z}_k^+$ and $\mathbf{Z}_k^-$ using a factor $2^k$.

We note that our experimental design ensures that every individual is assigned to treatment or control with equal probability. This implies that on expectation, the sizes of the treatment and control groups are equal (for every partitioning). However, when we consider a particular assignment, it is possible that the size of the smaller partition is not exactly half of the population. As a result, the total number of samples used might be smaller by a factor of at most 2.

**Theoretical Guarantees.** Our analysis approach, inspired by the coreset construction for discrepancy minimization [26], is based on the observation that if we can obtain good estimates for the contributions $\sum_{i \in [n]} \mathbf{y}_i^1$ and $\sum_{i \in [n]} \mathbf{y}_i^0$, we obtain a good estimate for ATE ($\tau$). Using the next lemma, we argue that after a call to GSW algorithm that partitions $[n]$ into the sets $\mathbf{S}^+$ and $\mathbf{S}^-$, we can obtain additive approximations of $\sum_{i \in [n]} \mathbf{y}_i^1$ and $\sum_{i \in [n]} \mathbf{y}_i^0$. Our approximations are the contributions of treatment and control values in $\mathbf{S}^+$ and $\mathbf{S}^-$ scaled appropriately, i.e., $\sum_{i \in \mathbf{S}^+} 2 \cdot \mathbf{y}_i^1$ and $\sum_{i \in \mathbf{S}^-} 2 \cdot \mathbf{y}_i^0$.

**Lemma 4.2.** *Suppose the Gram-Schmidt-Walk design [20] partitions the population $[n]$ into two disjoint groups $\mathbf{S}^+$ and $\mathbf{S}^-$. Under the linearity assumption, with probability $1 - 1/3 \log(n/s)$, for both the control and treatment groups, the following holds:*

$$\left| \sum_{j \in \mathbf{S}^+} 2\mathbf{y}_j^i - \sum_{j \in [n]} \mathbf{y}_j^i \right| \leq 4\sqrt{\log(16 \log(n/s))} \cdot \left( 2\sigma\sqrt{n} + \left\| \boldsymbol{\beta}^i \right\| \right) \quad \text{for } i = 0, 1.$$

Building upon the previous lemma, we argue in Theorem 4.3 that the additive approximation errors obtained from repeated use of GSW in our algorithm RECURSIVE-COVARIATE-BALANCING result in a low estimation error.

**Theorem 4.3** (Main ATE Error Bound). *The estimator $\widehat{\tau}_s$ in Algorithm* RECURSIVE-COVARIATE-BALANCING *obtains the following guarantee, with probability at least $2/3$:*

$$|\widehat{\tau}_s - \tau| = O\left( \sqrt{\log \log(n/s)} \cdot \left( \frac{\sigma}{\sqrt{s}} + \frac{\left\| \boldsymbol{\beta}^1 \right\| + \left\| \boldsymbol{\beta}^0 \right\|}{s} \right) \right).$$

**Remark.** When $s = n$, the above theorem matches the guarantees obtained by GSW design described in Lemma 4.1. Moreover, we obtain a better dependence compared to sampling $s$ rows uniformly at random and using the $\mathbf{y}^1, \mathbf{y}^0$ values of the sampled rows to estimate the population mean of treatment and control groups in ATE. An application of standard concentration inequalities or the central limit theorem, will yield a multiplicative factor increase in one of the error terms, with a dependence of $\widetilde{O}\left( 1/s \cdot \|\mathbf{X}\|_2 \left( \|\boldsymbol{\beta}^1\| + \|\boldsymbol{\beta}^0\| \right) \right)$, instead of the $\widetilde{O}\left( 1/s \cdot \left( \|\boldsymbol{\beta}^1\| + \|\boldsymbol{\beta}^0\| \right) \right)$ obtained by our algorithm, where $\|\mathbf{X}\|_2$ denotes the spectral norm of $\mathbf{X}$ and $\widetilde{O}(\cdot)$ hides the logarithmic factors.

## 5 Experimental Evaluation

In this section, we provide an evaluation of our algorithms on various semi-synthetic datasets. Missing details about data generation and additional results are collected in Appendix A.3.

**Data Generation.** We evaluate our approaches on five datasets: *(i) IHDP.* This contains data regarding the cognitive development of children, and consists of 747 samples with 25 covariates describing properties of the children and their mothers, and whose outcome values are simulated [21, 14]. *(ii) Twins.* This contains data regarding the mortality rate in twin births in the USA between 1989-1991 [5]. Following the work of [32], we select twins belonging to same-sex, with weight less

than 2kg, resulting in about 11984 twin pairs, each with 48 covariates. We use the post-treatment mortality outcomes of the twins as potential outcomes. *(iii) LaLonde.* This contains data regarding the effectiveness of a job training program on the real earnings of an individual after completion of the program [29], which is also the outcome value. The corresponding covariate matrix contains 445 rows and 10 covariates per row. *(iv) Boston.* This is constructed based on the housing prices in the Boston area [19]. The outcome value for each sample represents the median house price. The corresponding covariate matrix contains 506 rows and 12 covariates per row. *(v) Synthetic.* We construct a covariate matrix $\mathbf{X} \in \mathbb{R}^{2000 \times 25}$, using an approach due to [33]. There is a high disparity in leverage score values in $\mathbf{X}$, similar to what we observe in other datasets. Using a random linear function on $\mathbf{X}$ and adding Gaussian noise, we generate the potential outcomes.

For IHDP and Twins datasets, we use the simulated values for potential outcomes, similar to Shalit et al. [43] and Louizos et al. [32]. For the Synthetic dataset, we simulate values for the outcomes using linear functions of the covariate matrix. For Boston, Lalonde datasets, as we have access to only one of the outcome values, we chose to compare our algorithms for a fixed shift in treatment effect (i.e., the true treatment effect is equal to a constant), similar to Arbour et al. [6].

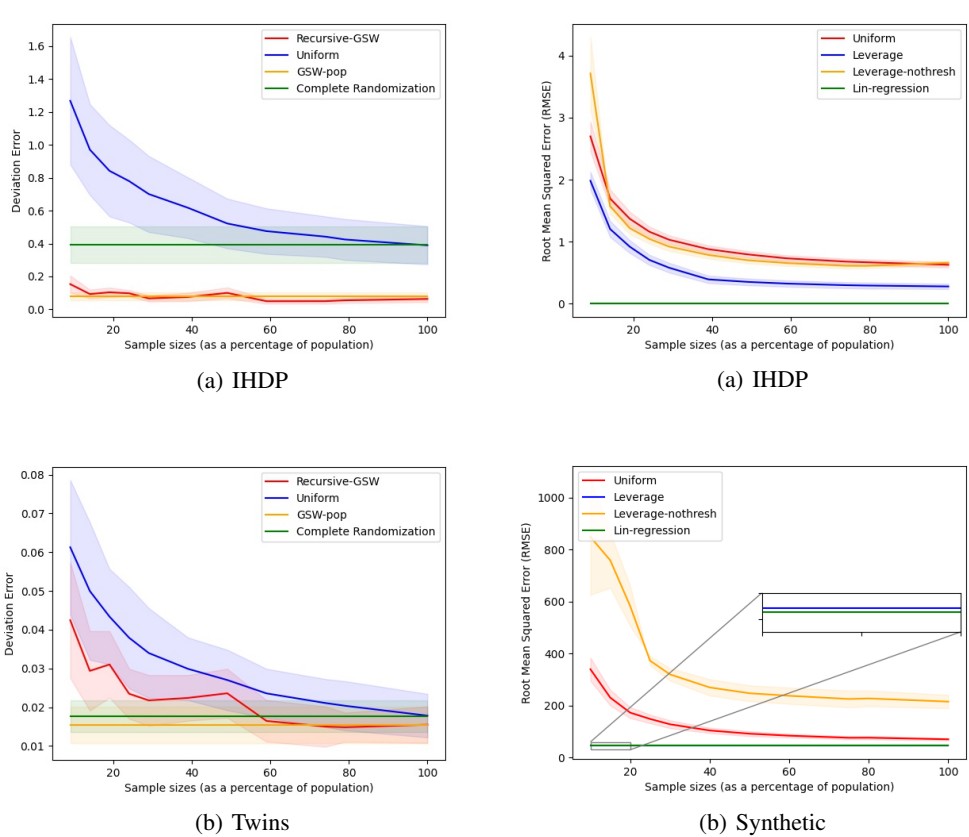

(a) IHDP       (a) IHDP

(b) Twins       (b) Synthetic

Figure 1: We compare the performance of various methods for estimating ATE, measured using deviation error on $y$-axis, against different sample sizes (as proportion of dataset size) on $x$-axis.

Figure 2: We compare the performance of various methods for estimating ITE, measured using RMSE on $y$-axis, against different sample sizes (as proportion of dataset size) on $x$-axis.

**Baselines.** (i) **ATE.** We compare the performance of our Algorithm RECURSIVE-COVARIATE-BALANCING (referred to as *'Recursive-GSW'*) to three baselines: *(i) Uniform.* We sample $s$ rows uniformly at random and assign them to treatment and control groups with equal probability. By scaling the total sum of treatment values from the sampled set by the inverse sampling probability, we estimate the contribution of treatment values in ATE and follow a similar procedure for the control

group. *(ii) GSW-pop.* We use the GSW algorithm to partition the full population and return the estimate obtained using the Horvitz-Thompson estimator for ATE. *(iii) Complete Randomization.* We partition the population into treatment and control using complete randomization, i.e., with equal probability, and return the estimate obtained using the Horvitz-Thompson estimator for ATE. The last two baselines are overall $n$ individuals rather than a subset of size $s$.

(ii) **ITE.** We compare the performance of our Algorithm SAMPLING-ITE (referred to as *'Leverage'*) with respect to three baselines: *(i) Uniform.* We run Algorithm 1 on $\mathbf{X}$ and uniform sampling distribution given by $\boldsymbol{\pi}_j = s/n \; \forall j.$ *(ii) Leverage-nothresh.* We run Algorithm 1 on $\mathbf{X}$, instead of $\mathbf{X}^*$ with the probability distribution $\boldsymbol{\pi}_j \propto \ell_j(\mathbf{X}) \forall j.$ *(iii) Lin-regression.* This captures the best linear fit regression error, i.e., assuming we have access to both $\mathbf{y}^1, \mathbf{y}^0$, we regress these vectors on $\mathbf{X}$ to obtain $\boldsymbol{\beta}^1, \boldsymbol{\beta}^0$, and use the resultant ITE estimates $\mathbf{X}\boldsymbol{\beta}^1 - \mathbf{X}\boldsymbol{\beta}^0$.

**Evaluation.** To evaluate the performance of average treatment effect estimation ($\tau$) on the datasets, we compare the deviation error of the estimator $\widehat{\tau}_s$, given by $|\widehat{\tau}_s - \tau|$ for different sample sizes. To evaluate the performance of individual treatment effect estimates, we compare the root mean squared error RMSE (see Defn. 2.4) for different sample sizes.

**Results.** For every dataset, we run each experiment for 1000 trials and plot the mean using a colored line. Also, we shade the region between 30 and 70 percentile around the mean to signify the *confidence interval* as shown in Figures 1, 2 representing ATE and ITE results respectively.

(i) **ATE**. For all datasets, we observe that the deviation error obtained by our algorithm labeled as *Recursive-GSW* in Figure 1, is significantly smaller than that of *Uniform* baseline. Surprisingly, for the IHDP dataset, our approach is significantly better than *Complete-randomization*, for all sample sizes, including using just 10% of data. For all the remaining datasets using a sample of size 30%, we achieve the same error (up to the confidence interval) as that of *Complete-randomization*. Complete randomization is one of the most commonly used methods for experimental design and our results indicate a substantial reduction in experimental costs. For IHDP dataset, a sample size of about 10% of the population is sufficient to achieve a similar error as that of *GSW-pop*. For the remaining datasets, we observe that for sample sizes of about 30% of the population, the deviation error obtained by our algorithm is within the shaded confidence interval of the error obtained by *GSW-pop*. Therefore, for a specified error tolerance level for ATE, we can reduce the associated experimental costs using just a small subset of the dataset using our algorithm.

(ii) **ITE**. For all sample sizes, we observe that the RMSE obtained by our algorithm labeled as *Leverage* in Figure 2, is significantly smaller than that of all the other baselines, including *Uniform* and *Leverage-nothresh*. E.g., we observe that when the sample size is 20% of the population in IHDP dataset, the error obtained by *Leverage* is at least 50% times smaller than that of *Uniform* and *Leverage-nothresh*. For the Synthetic dataset, the error obtained by *Leverage* is extremely close to that of the error due to the best linear fit, *Lin-regression* (see the zoomed in part of the figure). Similar to ATE results, our algorithms result in a reduction of experimental costs for ITE estimation using only a fraction of the dataset.

## 6 Conclusion

We study the sample constrained treatment effect estimation problem and give efficient algorithms for both ITE and ATE estimation. Our empirical evaluation shows that our algorithms, using only a fraction of the data, perform well compared to popular baselines that are widely used and require running experiments on the entire population. There are several interesting directions for future work. It would be interesting to study sample constrained treatment effect estimation under interference [47]. For ITE estimation, we leave it as an open question to extend our approach to give an algorithm with an error growing smaller with $s$ for all values of $s \leq n$. Moreover, the $\sqrt{\log d}$ factor in Corollary 3.7 likely can be improved, using recent work which improves standard leverage score sampling bounds by a $\log d$ factor [11], yielding a bound that is optimal up to constants. For ATE estimation, the Horvitz-Thompson estimator can include sampling probabilities different from $0.5$. It is an interesting open question to extend our recursive balancing approach when the estimator contains arbitrary probabilities.

## Acknowledgements

Most of this work was done while R. Addanki was a student at the University of Massachusetts Amherst and an intern at Adobe. Part of this work was done while R. Addanki was a visiting student at the Simons Institute for the Theory of Computing. This work was supported by a Dissertation Writing Fellowship awarded by the Manning College of Information and Computer Sciences, University of Massachusetts Amherst to R. Addanki. R. Addanki and C. Musco were additionally supported in part by NSF grants CCF-2046235, IIS-1763618, CCF-1934846, CCF-1908849, and CCF-1637536, along with Adobe and Google Research Grants.

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
