# OpenReview forum: "Sample Constrained Treatment Effect Estimation"
_NeurIPS.cc/2022/Conference — NeurIPS 2022 Accept_

### Official Review · Reviewer_Jafg · 2022-07-05

**Rating:** 7
**Confidence:** 2
**Soundness:** 3 good
**Presentation:** 2 fair
**Contribution:** 3 good

**Summary:**

This paper proposes two novel algorithms for designing efficient randomized controlled trials (RCT), so that they can be then used to estimate average and individual treatment effects. The algorithms provide ways to sample s << n points and assign them to treatment and control groups. The paper assumes linearity in the outcome functions, and shows both theoretically and empirically, that the proposed algorithms can design RCTs from which the treatment effects can be estimated accurately.


**Questions:**

- In assumption 2.1, it is stated that X is row-normalized. I can understand why we might want the design matrix to be standardized, but do not understand why it should be row-normalized.
- For the ITE estimation task, the samples are selected based on their leverage score (a sample with a higher leverage score has a higher chance of being selected). It is however not explained why these samples are more important than the other samples with lower leverage scores.
- In lines 158-160, it is stated that using a smoothed X would help mitigate sampling the same data point for both treatment arms. I don’t see how though.
- It is stated that \pi is calculated based on the leverage scores; could you please provide the exact formula for calculating \pi?
- Why is S^0 sampled according to \pi and S^1 according to \pi(1-\pi)? Is it because we want to reduce the number of treated subjects as much as possible? Can the reverse yield a viable RCT?
- Why are there two separate algorithms for designing RCTs, one for ATE and another for ITE? We can calculate accurate ATE from a dataset with which we can estimate accurate ITEs (I understand the reverse may not be true).


**Strengths And Weaknesses:**

The paper is addressing a very interesting and important problem. The findings in this paper can help design RCTs that are more efficient than the conventional method of randomization.
I however had a very hard time understanding a lot of the concepts discussed in the paper. I wished the paper had been written with more clarity.

---

> ### Author Response · Authors · 2022-08-01
> **Addressing Reviewer Jafg questions**
>
> 1. *" In assumption 2.1, ... "*
>
>     The row-normalization is to simplify the analysis and the final theorem statements. All of our results will hold without row normalization and our statements need to be updated with a dependence on the maximum row-norm of X. The final guarantees will have a dependence similar to the one described in Harshaw et al. [2019]. We will update the results in the final version.
>
> 2. *" For the ITE estimation ... "*
>
>     The leverage score of a row captures the relative importance of a row in comprising the row-space of any given matrix. E.g. the row with the highest leverage score might point to a direction orthogonal to all the other rows, capturing unique information that is required to find a near-optimal linear model. Therefore, the highest leverage score samples are essential as they help obtain accurate estimates of the ATE/ITE values. Rows with low leverage scores tend to align with many other rows that lie in similar directions. Thus, sampling just a small subset of these rows still allows us to learn the component of the linear model in their direction.
>
>
> 3. *" In lines 158-160, ... "*
>
>     Without smoothing, there is a possibility of high disparity in sampling probabilities, i.e., our experimental design could assign a particular treatment to an individual with a probability close to 1, while the remaining treatment would have an assignment probability closer to 0.  By smoothing the matrix, and ensuring the leverage scores are small, it is unlikely that an individual is sampled in both the treatment arms, i.e., S_0 and S_1. Our method also ensures that the sampling probability for both the treatment arms are separated only by a small constant factor (see Claim 3.4).
>
>
> 4. *" It is stated that $\pi$ ... "*
>
>     The value of $\pi = \min \( 1, \ell_j(X^*)  \cdot [c_0 \log (\mathrm{rank}(X^*)) + s/4 d\log d ] \)$, where $c_0$ is a constant and $\ell_j(X^*)$ is the leverage score of $j$th row of $X^*$ (see Definition 2.5). It is also described in Section 3.2.
>
>
> 5. *" Why is S^0 sampled according ... "*
>
>     The probability with which we include a row in $S^1$ is simply the probability that it is sampled ($\pi$), but not also included in $S^0$ ($(1-\pi)$). Multiplying both the values gives us a total probability of $\pi (1-\pi)$. As the reviewer correctly points out it is possible to remove the repeated samples from $S^0$, instead of $S^1$, resulting in similar probability expressions, or drop the common samples randomly from either $S^0$ or $S^1$. For all such cases, we can obtain the same guarantees as that of Theorem 3.6. We will clarify this in the final version.
>
> 6. *" Why are there two separate algorithms ... "*
>
>     ITE estimation requires accurate estimates for each individual, whereas, with ATE estimation, we only need to estimate the average of all the ITE values. Estimating ATE by first estimating ITE values, might result in weaker estimates – E.g., by comparing the RMSE for ITE (Thm 3.6)  and deviation error bounds for ATE (Thm 4.3), we can observe that estimating ATE using ITE estimates obtained from Section 3 results in a significantly higher error compared to our approach in Section 4.

---

> > ### Comment · Reviewer_Jafg · 2022-08-06
> > **After reading the rebuttal**
> >
> > I would like to thank the authors for their rebuttal.
> > As they have addressed all my concerns, I have increased my score.

---

### Official Review · Reviewer_ubt5 · 2022-07-07

**Rating:** 7
**Confidence:** 3
**Soundness:** 3 good
**Presentation:** 4 excellent
**Contribution:** 3 good

**Summary:**

This paper proposed methods to estimate individual treatment effects (ITEs) and average treatment effects (ATEs) under a budget constraint and a linear model assumption on the potential outcomes. The method for estimating ITEs is based on the idea of leverage score  sampling [51] in numerical algebra. The authors apply the idea to sample a limited set of observed vectors to span the column space of the design matrix in the linear regression models for the treated and control outcomes. The authors then extend their method to estimate ATE, building upon the recent Gram-Schmidt work [20] on covariate balancing in experimental design. The authors provide theoretical guarantees and experimental results for their methods in both settings.



**Questions:**

In line 184, if one row gets included in both $S^0$ and $S^1$, the authors remove the row from $S^1$. Why not further randomize this with 1/2 and 1/2 probabilities for removing the row in  $S^0$ and $S^1$, respectively.  In your proposal, we are expected to end up with less treated samples.

**Ethics Review Area:**

["I don’t know"]

**Strengths And Weaknesses:**

Strengths:
1. I believe the problem studied in this paper is important for treatment effect estimation.
2. The method based on leverage score is an interesting and good contribution to the field.
3. The theoretical results under a linear model give good guarantees for the method's performance.

Weaknesses:
1. The definition 2.2 of ITE is correct, but the return of Algorithm 1 is $\mathbf X^* \beta_1 - \mathbf X^* \beta_0$, which is the conditional average treatment effect, i.e. the expectation of the ITE conditional on X. In other words, the noise in the outcome goes to 0 after taking the expectation. In fact, the ITE for a given individual is a random variable rather than a parameter, so does take a fixed value. So we can not estimate it exactly, instead, methods should provide a prediction interval to quantify the randomness in ITE.
2. I don't fully understand the transition from Lemma 3.3 (Follows from [41]) to Theorem 3.6. First Is $S = S^0\cup S^1$ in Theorem 3.6? I thought this is the case after reading the proof in Appendix. I thought the main idea is still the same about controlling the expectation of $|S|$. Could you explain what has been designed differently for causal inference in which we only observe one outcome for every individual? I feel like the difference is just adding the step with pi(1-pi) probability in Algorithm 1
3. It is interesting to see a theoretical guarantee on estimation errors. It is unclear to me whether this method still allows users to do
finite population inference for the treatment effects, i.e.,  construct valid confidence intervals for the treatment effects.
4. Section 4 is not very easy to understand if readers (like me) don't know the Gram-Schmidt-Walk design very well. Assuming a linear model seems like a limitation when working with the Horvitz-Thompson Estimator.

---

> ### Author Response · Authors · 2022-08-01
> **Addressing Reviewer ubt5 questions**
>
> 1. *" The definition 2.2 of ITE is correct..."*
>
>     (a) We consider the finite population set up, with a linearity model assumption between potential outcome values and a fixed set of covariates X, where the source of randomness is the design of the experiment and noise in the model. The reviewer suggests a slightly different setting from ours, with a distributional assumption on the covariates. Although we believe the settings might not be directly comparable, such a setting could be an interesting direction for future work.
>
>     (b) Using Thm 3.6, we argue that the root-mean-squared-error of our ITE estimates is small, and clearly distinguishes the contribution of both the mentioned sources in the error guarantees. Finally, we argue in Cor 3.7 that such a bound is optimal. Our guarantees can be interpreted as bounding the confidence interval around the root-mean-squared error for a fixed failure probability. We will update our error guarantees that include a general failure probability of $\delta$, which is probably what the reviewer was suggesting.
>
>
> 2. *" I don't fully understand..."*
>
>     (a) Yes, as the reviewer correctly points out, we include both treatment and control samples, i.e., $S = S^0 U S^1$ in Theorem 3.6.
>
>     (b) The main difference, as we clarify in the introduction, is the fact that we need to solve two simultaneous linear regression problems, while only having access to one of the labels for every data point. The main difficulty is in ensuring that a particular point is not included in both $S^0$ and $S^1$, as we have access to only one of the treatment outcomes. We overcome this by including points that are sampled in both $S^0$ and $S^1$, by dropping them from $S^1$, resulting in a probability of $\pi (1-\pi)$. The probability with which we include a row in $S^1$ is simply the probability that it is sampled (denoted by $\pi$), but not also included in $S^0$ (denoted by $(1-\pi)$). Multiplying both the values gives us a total probability of $\pi (1-\pi)$.
>
>
> 3. *" It is interesting to see ..."*
>
>     Our guarantee shows that our ITE estimate's root-mean-squared error is bounded.  Although such an error guarantee might not directly translate to confidence interval bounds for individual treatment effects, we want to highlight such error estimates have been previously studied (see Shalit et al. [2017]). We agree with the reviewer that confidence intervals for each ITE estimate would be an interesting direction for future work.
>
>
> 4. *" Section 4 is not very easy ... "*
>
>     (a) Gram-Schmidt-Walk design is a recently proposed algorithm that constructs a partition of the population, ensuring that the covariates across the partition (i.e., treatment and control group) are very similar. The choice of the Horvitz-Thompson estimator is not because of the linearity assumption, as the Horvitz-Thompson Estimator has been well-studied for both linear and non-linear models (see Harshaw et al. [2019] for an extensive discussion).
>
>     (b) Although the Gram-Schmidt-Walk design doesn’t make the linearity assumption, the error guarantee depends on the best linear fit and therefore is robust against model misspecification. We make the linearity assumption to simplify the analysis and a similar guarantee that is robust against model misspecification can possibly be obtained with more careful analysis.
>
>
> 5. *"In line 184, if one row..."*
>
>     Yes, the reviewer makes a good observation. Such an approach will also work and results in a claim same as Theorem 3.6. We will include a clarifying statement in the final version.

---

### Official Review · Reviewer_8nKe · 2022-07-08

**Rating:** 5
**Confidence:** 4
**Soundness:** 2 fair
**Presentation:** 2 fair
**Contribution:** 3 good

**Summary:**

The paper propose novel algorithm to estimate ITE and ATE, in the setup where the budget of targeted persons is constraints. Namely we can only touch s<<n persons, and would like to select these s people to estimate the ITE and ATE the best as possible.

**Questions:**

- All bounds in probability are or fixed delta, like delta=1/30. why not letting delta unfixed, so that we can check some limit cases, and then expliciting the same result for delta=1/30?
- the choice of removing j from S^1 if j \ in S^0 and S^1 sound odd. why so?
- the case where n=2^k -1 and s=2^(k-1) wil only use s'=2^(k-2) samples, and "waste" s-s' samples if I am right?
- I am not convinced by the explainations on line 234-235. The empirical analysis could be more various to assess this
- the Horvitz-Thomp!son estimator could be more general, with sampling probablility explicited: A/pi-B/(1-pi). The factor 2 looks very unexplained, and the work could be easily generalized to other pi
- Remark line 284 is nonsense, if n=s, then previous bound is -infinity
- code is not working, and would deserve a more precise readme file: "Requires installing python3, julia and interfacing it with other open source code mentioned in the paper."
-  proof Th A9, does not hold of n is not 2^k
-  30-70% percentile looks not standard at all why so?
-  all plots are not vectorised: difficult to read them

typos:
line 54 They->The
line 55 both select-> select both
line 118 each-> each of them
line 184 simple-> simply
line 242 that: bad formulation

**Limitations:**

ITE and ATE are deeply linked with privacy of data and could lead to privacy leak, especially for ITE.

**Strengths And Weaknesses:**

originality:
this is one of the first approach using other tools such as leverage score sampling, in a potential outcome setup, willing to estimate ITE and ATE.

quality:
the mathematical sections are roughly interesting, and the theorem+corrolary are easy to follow. All definitions are clear.

clarity:
the paper is roughly smooth to read, although there could be incremental improvement

significance:
the significance of such sampling constraints could be better motivated, with various concrete example.

---

> ### Author Response · Authors · 2022-08-01
> **Addressing Reviewer 8nKe questions**
>
> 1. *"All bounds in probability are or fixed delta..."*
>
>     We thank the reviewer for the suggestion. For the sake of simplicity of analysis, we used a constant success probability. All our claims can easily be updated with a general failure probability of $\delta$, with a dependence of $1/\delta$ (similar to Lemma 3.3). We will update the statements in the final version.
>
>
> 2. *" the choice of removing..."*
>
>       If we include $j$ in both $S^1$ and $S^0$, it is equivalent to accessing both $Y^0$ and $Y^1$ values for the row $j$, which is not allowed. We could remove $j$ randomly from one of $S^0$ or $S^1$ and obtain the exact same guarantees -- for simplicity of the analysis, we remove it from $S^1$.
>
>
> 3. *" the case where $n=2^k -1$..."*
>
>      When $n = 2^k, s = 2^{k-1} - 1$, we will use $s' = 2^{k-2} - 1$ samples. Our experimental design ensures that every individual is assigned to treatment or control with equal probability. This implies that on expectation, the sizes of the treatment and control groups are equal (for every partitioning). However, when we consider a particular design, it is possible that the size of the smaller partition is not exactly half of the population. As a result, the total number of samples used might be smaller by a factor of at most 2, as pointed out by the reviewer.
>
>
>
> 4. *" the Horvitz-Thompson estimator could be more general..."*
>
>      We agree with the reviewer that the Horvitz-Thompson estimator is more general and can include different sampling probabilities, the version where $\pi = \frac{1}{2}$ has been well studied. We will clarify this.  It is an interesting open question to extend our recursive balancing approach when the estimator contains arbitrary probabilities.
>
>
> 5. *" Remark line 284 is ..."*
>
>      We disagree with the reviewer. If n = s, we would be assigning the population to control and treatment groups using the Gram-Schmidt-Walk design, and we achieve a bound of $O\left( \frac{\sigma}{\sqrt{n}} +  \frac{\left\lVert \pmb{\beta}^1 \right\rVert + \left\lVert \pmb{\beta}^0 \right\rVert }{n} \right).$ We can observe that the previous bound provided by Harshaw et al. [2019] would follow in the case $ n = s$.
>
> 6. *" ... precise readme file ... "*
>
>     We will update the readme file with precise instructions on how to run it. In particular, it requires open source code for Gram-Schmidt-Walk design (Harshaw et al.[2019]) and Julia packages.
>
> 7. *" proof Th A9, does not ..."*
>
>     Proof of Thm A9 does not assume that $n$ is $2^k$. It merely recurses over the smaller of the two partitions at every level. The terms corresponding to $n/2^k, n/2^{k-1}, ...$ in the proof correspond to upper bounds (and not the exact sizes) on sizes of intermediate partitions created by our recursive partitioning approach.

---

### Meta-Review · Area_Chair_QzmK · 2022-08-24

**Recommendation:** Accept
**Confidence:** Certain

**Metareview:**

The reviewers came to consensus that this paper makes a good progress on the study of treatment effect estimation. I agree with these opinions and please polish the manuscript so that the concerns raised by the reviewers become clear in the final version. In particular, I'm also interested in the concern on the constant $\delta$ and I expect that this point is addressed so that the dependence of the results on delta becomes clear.

**Award:**

No

---

### Decision · Program_Chairs · 2022-09-14

Accept